# MOTION MARIONETTE: RETHINKING RIGID MOTION TRANSFER VIA PRIOR GUIDANCE

## ABSTRACT

We present *Motion Marionette*, a zero-shot framework for rigid motion transfer from monocular source videos to single-view target images. Previous works typically employ geometric, generative, or simulation priors to guide the transfer process, but these external priors introduce auxiliary constraints that lead to trade-offs between generalizability and temporal consistency. To address these limitations, we propose guiding the motion transfer process through an internal prior that exclusively captures the spatial-temporal transformations and is shared between the source video and any transferred target video. Specifically, we first lift both the source video and the target image into a unified 3D representation space. Motion trajectories are then extracted from the source video to construct a spatial-temporal (SpaT) prior that is independent of object geometry and semantics, encoding relative spatial variations over time. This prior is further integrated with the target object to synthesize a controllable velocity field, which is subsequently refined using Position-Based Dynamics to mitigate artifacts and enhance visual coherence. The resulting velocity field can be flexibly employed for efficient video production. Empirical results demonstrate that Motion Marionette generalizes across diverse objects, produces temporally consistent videos that align well with the source motion, and supports controllable video generation. Demo videos are available in this anonymous project page.

## 1 INTRODUCTION

Motion transfer, the task of transferring motion patterns from videos to static images, has attracted considerable attention across diverse areas, including content creation (Wang et al., 2024c; Li et al., 2024c; Geng et al., 2024), augmented and virtual reality (AR/VR) (Sun et al., 2023; Aberman et al., 2020; Wang et al., 2025), and robot state prediction (Zhang et al., 2024c; Li et al., 2024b). Despite its potential, broader applicability remains constrained by two fundamental challenges. Firstly, accurately capturing and modeling the complex motion dynamics inherent in source videos is a non-trivial task. Secondly, effectively adapting these extracted motion patterns onto static target images to generate coherent and visually consistent videos remains difficult. To date, neither of these challenges has been sufficiently addressed, highlighting the need for further explorations.

Researchers have been attempting to circumvent the two critical challenges by introducing different priors. One line of studies focuses on transferring motion to images that depict subjects within the same category as those in the source videos. These approaches (Sun et al., 2023; Wang et al., 2025) typically leverage robust *geometric priors* associated with specific object categories (e.g., human bodies and faces), utilizing parametric models to facilitate accurate shape and pose transformations. Another prominent research direction involves diffusion-based video editing methods (Wu et al., 2023; Geyer et al., 2023), which integrate *generative priors* to replace or modify the original subjects while preserving identical motion dynamics (Yatim et al., 2023; Meral et al., 2024; Jeong et al., 2024). Additionally, physics-based methods (Xie et al., 2023; Tan et al., 2024; Fu et al., 2024), employing *simulation priors*, have recently gained popularity due to their strong capability in realistically modeling dynamic and deformable objects.

Although these methods have produced visually compelling results, their reliance on the adopted *external* priors introduces significant trade-offs. Approaches utilizing parametric models inherently constrain the variety of motions and object types they can accommodate, resulting in limited general-

izability. Meanwhile, generative-based editing methods inherit the intrinsic limitations of diffusion models. While capable of generating diverse outputs, they frequently struggle with maintaining shape integrity and temporal consistency. Additionally, simulation-based techniques heavily depend on strict assumptions about object materials and manually specified physical rules, which often fall short in capturing the variability and complexity of real-world scenarios.

We observe that the above limitations ultimately stem from the auxiliary constraints introduced by the incorporation of external priors, which impose assumptions unrelated to the core motion transfer task. In light of this, we **rethink** motion transfer as a process focused exclusively on transferring spatial variations over time. To avoid incorporating extraneous assumptions and to enhance generalizability, this process should be guided by an *internal* prior that captures spatial-temporal transformations while remaining independent of object category and absolute spatial position. We define this as the ***spatial-temporal (SpaT) prior***, a shared motion representation between the source and any transferred target videos. Our objective, therefore, is to construct and leverage this SpaT prior to facilitate generalizable, coherent, and computationally efficient motion transfer.

To this end, we introduce Motion Marionette, a novel paradigm specifically designed for rigid motion transfer—including translation, rotation, and oscillation—from monocular videos onto single-view static images. Our pipeline first lifts both the source video and target image into a unified 3D representation space. Subsequently, we extract motion trajectories from the source video to construct a robust SpaT prior, effectively capturing rigid relative spatial transformations over time. This generalizable prior is shared by the static target object to construct an explicit velocity field, from which motion is synthesized via Euler integration steps. To improve compatibility between the velocity field and the target object's geometry, we employ an iterative refinement procedure inspired by Position-Based Dynamics (PBD) (Müller et al., 2007), reducing the accumulated errors during Euler integration over long sequences. Finally, the explicit velocity field enables efficient rendering of the transferred video and can be flexibly manipulated to support controllable video generation with diverse motion dynamics and camera viewpoints.

To validate the effectiveness of Motion Marionette, we facilitate open-source video datasets with high-quality image generation tools for method evaluation, demonstrating its ability to produce videos with consistent motion and strong temporal coherence. Furthermore, ablation studies reveal that our approach generalizes well across diverse object types and supports the generation of an arbitrary number of videos with varying motion speeds and camera poses. These results underscore the potential of Motion Marionette for accurate motion transfer and controllable video generation.

## 2 RELATED WORKS

**Motion Transfer from Videos.** Transferring motion dynamics from a source video to target objects is compelling. Much of the existing work focuses on transferring motion between subjects of similar categories or with comparable structural properties, particularly human faces and bodies (Sun et al., 2023; Chen et al., 2023a; Aberman et al., 2020; Wang et al., 2025; Siarohin et al., 2019; Maheshwari et al., 2023; Zhang et al., 2025a). By directly leveraging or learning a pre-defined parametric model, these methods align the source video and target object features over time for effective motion transfer. Another series of works, which can also be seen as a variation of video editing, focused on enhancing the power of different generative models (Shi et al., 2025; Jeong et al., 2023; Park et al., 2024a; Ren et al., 2024; Wang et al., 2024a; Zhao et al., 2023; Yatim et al., 2023; Wu et al., 2023; Geyer et al., 2023; Meral et al., 2024) to directly generate a video with transferred motion. These methods implicitly encode motion patterns from the source video and align intermediate features, often with the aid of text input, to guide the output. A less relevant line of works is an extension of 4D generation from video inputs (Jiang et al., 2024; Zeng et al., 2024; Wu et al., 2024; Li et al., 2024e; Zhang et al., 2024a; Yang et al., 2025). Based on Score Distillation Sampling (Poole et al., 2022), these methods adopt video diffusion models to generate view-consistent dynamic objects. Motions are encoded implicitly as latent features and can be applied onto a new input to realize motion transfer. However, the transferability of the motions are also limited (Wu et al., 2024), where the target object's structure need to be carefully aligned with the generated object.

**Data-driven Simulation.** An emerging direction in 3D vision is to integrate physics-based simulation tools (Liu et al., 2025), with the Material Point Method (Hu et al., 2018) being particularly

prominent. This simulation system allows for topology changes and frictional interactions, thus is especially useful for dealing with deformable objects and subject interactions. By assuming specific material properties and physical laws, these works (Xie et al., 2023; Tan et al., 2024; Borycki et al., 2024; Lin et al., 2025) enable novel motion synthesis, predicting object movements and interactions with the environment. Fu et al. (2024) further added video guidance for motion transfer, using the simulation environment to learn mappings between motions and deformable objects.

**Video Generation with Motion Guidance.** Recent progress in image generation has catalyzed advancements in video generation, particularly under text- and image-conditioned settings (Brooks et al., 2024; WanTeam et al., 2025; Zhang & Agrawala, 2025). Given the dynamic nature of videos, an increasing body of research has focused on motion-guided video generation, which leverages motion prompts to achieve more precise and controllable temporal synthesis. Early works in motion-guided video generation focused on using sparse motion cues (Hao et al., 2018; Ardino et al., 2021). More recent works extended this paradigm to utilize more complex motion trajectories (Wang et al., 2024c; Chen et al., 2023b; Li et al., 2024d; Mou et al., 2024). For instance, Chen et al. (2023b); Yin et al. (2023); Zhang et al. (2025b) generate videos conditioned on sparse motion trajectories, while Geng et al. (2024) employs dense motion tracks for finer control. Additionally, Wang et al. (2024c); Li et al. (2024c) incorporate camera trajectories to enhance the realism and expressiveness of generated videos. Another relevant line of works (Li et al., 2024a;b) focused on robotics, generating subsequent states and motions for the given image or 3D input and motion direction.

Our work differs from these prior studies in three aspects: (1) Instead of relying on parametric models and generative priors, we propose a simulation-free paradigm that facilitate the spatial-temporal prior for motion transfer; (2) We aim for rigid movements and does not focus on deformable objects, tackling translation, rotation and oscillation; (3) Motions are extracted and processed explicitly, enabling improved interpretability for effective transfer and flexible control for generation.

## 3 PRELIMINARIES

### 3.1 PROBLEM FORMULATION

The inherent complexity of motion transfer has led researchers to explore a variety of problem settings. To clearly define our scope, we formalize our task as follows: Given a source monocular video containing a moving object $X$ and a single arbitrary target image depicting a different object $Y$, the objective is to transfer the motion patterns exhibited by $X$ onto $Y$ without relying on generative models or physics-based simulation tools. The desired output is a coherent and visually realistic video sequence in which object $Y$ exhibits motion consistent with that of $X$. As an initial exploration of this task, we aim to achieve approximate transformations that preserve the semantic consistency of the motion. That is, the motion in the generated video should remain perceptually aligned with the dynamics exhibited in the source video.

### 3.2 3D RECONSTRUCTION FROM A SINGLE IMAGE

To recover 3D scenes from 2D observations, works such as Neural Radiance Fields (NeRF) (Mildenhall et al., 2020) and 3D Gaussian Splatting (3DGS) (Kerbl et al., 2023) have shown remarkable progress, enabling accurate reconstruction and improved interpretation of static 3D scenes from dense multi-view imagery. Among these paradigms, 3DGS, which is based on anisotropic spherical Gaussians, offers explicit and interpretable scene representations and surpasses NeRF in terms of both training and rendering efficiency. Given these advantages, we adopt 3DGS to represent all objects throughout this work. However, reconstructing 3D representations from a single view currently remains a significant challenge (Smart et al., 2024). Therefore, we **do not aim for perfect reconstruction fidelity**, as our primary focus is motion transfer rather than high-quality reconstruction.

### 3.3 MOTION TRAJECTORIES FROM MONOCULAR VIDEOS

We represent the motion trajectories as a set of long-range 3D trajectories of scene points over $T$ time steps, denoted as $\mathcal{T}^k = \{\boldsymbol{\tau}_t^k\}_{t=1}^T$, where $\boldsymbol{\tau}_t^k \in \mathbb{R}^3$ denotes the 3D position of the $k$-th trajectory at time $t$. To recover these trajectories from monocular video input, we utilize metric depth

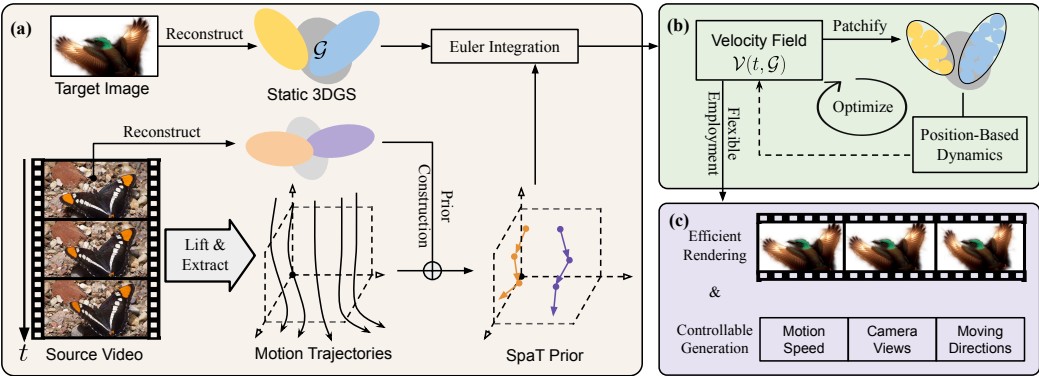

Figure 1: **Overview of Motion Marionette.** **(a)** We lift both the source video and the target image into 3DGS representations. Motion trajectories are then extracted from the source video and used to construct the SpaT prior, which is integrated with the target object using Euler integration to produce a velocity field that guides motion transfer. **(b)** We patchify the velocity field and perform iterative optimization to mitigate error accumulation caused by the absence of supervision and the use of Euler integration. **(c)** The explicit velocity field can thus be flexibly utilized for efficient rendering of coherent videos and also enables controllable video generation.

estimators (Hu et al., 2024a;b; Piccinelli et al., 2024) to predict per-frame depth maps $d_t : \mathbb{R}^2 \to \mathbb{R}_+$, and pixel trackers (Doersch et al., 2024; Karaev et al., 2024; Xiao et al., 2024) to extract 2D trajectories $\{\mathbf{u}_t^k\}_{t=1}^T$, where $\mathbf{u}_t^k \in \mathbb{R}^2$ denotes the pixel location of the $k$-th point in frame $t$. Following standard practice (Stearns et al., 2024; Wang et al., 2024b; Lei et al., 2024; Park et al., 2024b; Liang et al., 2025; Liu et al., 2024), we lift each 2D trajectory into 3D world coordinates using:

$$\boldsymbol{\tau}_t^k = \mathbf{W}_t \pi_{\mathbf{K}}^{-1} \left( \mathbf{u}_t^k, d_t(\mathbf{u}_t^k) \right), \tag{1}$$

where $\pi_{\mathbf{K}}(\cdot)$ denotes the projection function from camera space to image space given intrinsic parameters $\mathbf{K}$, and $\mathbf{W}_t$ the estimated camera pose at frame $t$. Note that the camera parameters are estimated using bundle adjustment (Lei et al., 2024) when not provided.

## 4 METHODOLOGY

This section details the key components of Motion Marionette. First, we describe the construction process of the spatial-temporal prior. Next, we explain how this prior is integrated with the target image to perform motion transfer. Finally, we demonstrate how Motion Marionette supports controllable video generation. An overview of the complete pipeline is provided in Fig. 1.

### 4.1 SPATIAL-TEMPORAL PRIOR CONSTRUCTION

Given a monocular input video consisting of $T$ frames, we construct the spatial-temporal (SpaT) prior through a two-stage process. First, we extract 3D object motion trajectories from the video. Then, we construct the SpaT prior under the assumption of rigid motion constraints.

**Object Motion Trajectory Extraction:** Given an input monocular video, we extract motion trajectories as described in Sec. 3.3. Unlike prior works (Lei et al., 2024; Stearns et al., 2024) that sample a sparse set of trajectories, we uniformly and compactly sample the scene in 3D space to construct a dense trajectory set $\mathcal{T} = \{\mathcal{T}^k\}_{k=1}^K$ of size $K$. This dense sampling strategy ensures broader spatial coverage, producing more informative motion representations for subsequent prior estimation.

However, the sampled trajectory set $\mathcal{T}$ includes trajectories corresponding to background regions, which are irrelevant for motion transfer. To isolate foreground motion, we project the 3D trajectories onto the 2D image plane (after scene normalization via scaling), and obtain per-frame foreground masks $\mathbf{M}_t$ using a segmentation model (Zheng et al., 2024; Ravi et al., 2024). These masks are then applied to the trajectory set in a time-consistent manner to retain only the relevant foreground trajectories. To preserve important boundary trajectories that may otherwise be lost due to projection artifacts, we employ a sliding-window masking strategy across the temporal domain. The final

trajectory set is constructed as a union of all masked trajectories across time:

$$\widetilde{\mathcal{T}} = \bigcup_{t=1}^{T} \mathbf{M}_t(\mathcal{T}). \tag{2}$$

This trajectory set therefore serves as an informative initialization for our SpaT prior. One may ask why not directly extract object motion trajectories from a masked video. The reason is that we find an empty background can be misleading for depth estimation and trajectory calculation.

**SpaT Prior Construction:** Although the extracted motion trajectories $\widetilde{\mathcal{T}}$ contain rich information, they are highly sensitive to the source object's position, geometry, and scale. As a result, directly applying these trajectories to the target image often leads to incorrect motion–object correspondence, due to mismatched spatial structures and scale disparities.

To ensure that the extracted motion is generalizable and position-independent, we aim to obtain a robust representation of relative spatial changes over time. To achieve this, we first reconstruct a 3DGS of the source object using the first frame of the source video, containing $N_s$ points. For each rigid motion component, we extrapolate the corresponding extracted motion trajectory across the rigid region so that it exhibits motion behavior consistent with the source video. Then we compute the rigid transformation between consecutive time steps using a least-squares alignment approach following Umeyama's method (Umeyama, 1991). Specifically, let $\mu_{t,i}^s$ and $\mu_{t+1,i}^s$ denote the $i$-th source 3D Gaussian's center positions at two consecutive time steps, we compute the cross-covariance matrix:

$$\mathbf{H}_t = \sum_{i=1}^{N_s} (\mu_{t,i}^s - \bar{\mu}_t^s)(\mu_{t+1,i}^s - \bar{\mu}_{t+1}^s)^\top, \tag{3}$$

where $\bar{\mu}_t^s$ and $\bar{\mu}_{t+1}^s$ are the centroids of the two Gaussian sets. We then perform singular value decomposition $\mathbf{H}_t = \mathrm{U}\Sigma\mathrm{V}^\top$, and compute the optimal rotation and translation:

$$\mathbf{R}_t = \mathrm{V}\,\mathrm{diag}\big(1,\ldots,1,\det(\mathrm{VU}^\top)\big)\,\mathrm{U}^\top, \quad \boldsymbol{\delta}_t = \bar{\mu}_{t+1}^s - \mathbf{R}_t\bar{\mu}_t^s. \tag{4}$$

Here, $\mathbf{R}_t \in \mathrm{SO}(3)$ and $\boldsymbol{\delta}_t \in \mathbb{R}^3$ describe the rigid transformation between time step $t$ and $t+1$. This transformation can be directly applied to any target point set $\boldsymbol{\mu}_t^d$ via:

$$\boldsymbol{\mu}_{t+1}^d = \mathbf{R}_t\boldsymbol{\mu}_t^d + \boldsymbol{\delta}_t. \tag{5}$$

By storing the sequence of $\mathbf{R}_t$ and $\boldsymbol{\delta}_t$ across all time steps, we define the SpaT prior, which serves as a robust motion descriptor for subsequent transfer to the target image.

## 4.2 MOTION TRANSFER WITH PRIOR GUIDANCE

Given the SpaT prior, which encodes transferrable rigid motions, and the reconstructed 3D Gaussian Splatting (3DGS) representation of the target object $\mathcal{G}$ from a single image, we compute the velocity field $\mathcal{V}(t, \mathcal{G}) = \{\boldsymbol{v}_t\}_{t=1}^{T-1}$ using Eq. 5. This velocity field defines the temporal motion of the target object in 3D space. Then the position of each Gaussian center at time $t+1$ is then updated via a simple Euler integration step:

$$\boldsymbol{\mu}_{t+1} = \boldsymbol{\mu}_t + \boldsymbol{v}_t, \tag{6}$$

where $t \in [1, T-1]$ and $\boldsymbol{\mu}_t$ denotes the positions of the Gaussians at time step $t$.

Due to the absence of supervision during zero-shot motion transfer, prediction errors arise and may accumulate over time. One observable artifact is the occurrence of gradual structural separations, caused by misaligned velocity directions over extended motion sequences. Another arises in scenarios involving abrupt spatial variations in motion (such as the moving wings and static body of a butterfly), where disconnectivity may occur due to insufficient geometric continuity across motion components. We next present strategies aimed at addressing these challenges.

**Kinematic Refinement:** Variations in object geometry can introduce subtle deviations in velocity directions, resulting in gradually separating structures. To address this, we first apply a *patchification* strategy to the 3DGS representation, dividing the target object into spatially local regions to enable efficient and localized optimization. A local connectivity graph is constructed using a compact KD-Tree (Bentley, 1975), allowing for fast retrieval of neighboring points. Thus, we can achieve the neighbor point set $\mathcal{N}(i)$ for any point $i$ in the target image's 3DGS representation with $N$ ellipsoids.

Due to the accumulation of errors in the Euler steps and the unstructured 3DGS distribution, artifacts gradually appear and become significant, which is reflected via small gaps appearing within originally compact regions. Drawing inspiration from Position-Based Dynamics (Müller et al., 2007), which aims to alleviate the integration of force information and enforces local geometry constraints through the space distribution itself, we perform Jacobi sweeps over the velocity field. Nevertheless, refining the velocity field of time $t$ does not consider the change of velocities over time, which may produce abrupt motion changes. To reduce the high-frequency noises or discontinuities after velocity optimization, we further adjust the acceleration term. Therefore, the kinematic loss is formulated as:

$$\mathcal{L}_{\text{kin}}^t = \sum_{i=1}^{N} \sum_{j \in \mathcal{N}(i)} \left[ \mathbf{1}(t < T) \left\| v_{t,i} - v_{t,j} \right\|_2^2 + \mathbf{1}(t < T-1) \left\| a_{t,i} - a_{t,j} \right\|_2^2 \right], \tag{7}$$

where $a_{t,i} = v_{t+1,i} - v_{t,i}$ is the acceleration of point $i$ at frame $t$, $\mathbf{1}(\cdot)$ is the indicator function.

**Topological Smoothing:** When the motion in the source video involves multiple rigid transformations, the target object correspondingly inherits distinct individual velocity fields. We observe that combining these fields often results in disconnectivity within the target object, causing it to appear fragmented during movements. An intuitive solution would be to establish an accurate point-wise mapping between the source and target objects, such that the relative spatial changes can be preserved. However, in most cases, the geometric differences between the source subject and the target object make even approximate point-wise mappings infeasible.

Instead of finding mappings, we focus on improving the topological relationships between the disconnected parts. Firstly, approximate motion boundaries are achieved by performing graph flood-fills starting from certain seed points, where the seed points can be obtained by human annotation or semantic matching (Zhang et al., 2024b). Then we iterate over the Gaussians in the motion boundaries to optimize the local velocity field such that it flows smoothly, avoiding potential spiking changes. For the boundary consisting of $M$ points, the topological loss for updating $\mathcal{V}$ at time $t$ is:

$$\mathcal{L}_{\text{topo}}^t = \frac{1}{M} \sum_{i=1}^{M} \left\| v_{t,i} - \frac{1}{|\mathcal{N}(i)|} \sum_{j \in \mathcal{N}(i)} v_{t,j} \right\|_2^2. \tag{8}$$

**Motion Propagation:** Although the above optimization strategies applied to the velocity field are effective in repairing structural separations and resolving disconnectivity, they may introduce self-collisions. This often manifests as discrepancies within intuitively static regions, where a subset of points $S_r$ remains stationary while adjacent areas $S_d$ become inadvertently dynamic, resulting in visually inconsistent transitions. To mitigate this abruptness, we propose to diffuse "pseudo" velocities into the static areas to preserve local rigidity and ensure smooth temporal evolution. Specifically, we propagate the mean velocity of the dynamic set $S_d$ to all points in $S_r$, ensuring consistent motion flow and perceptual coherence across time steps. The static set is defined as $S_r = \{i : \|v_{t,i}\| < \epsilon\}$, where $\epsilon$ is a predefined small threshold for motion filtering.

Integrating the above strategies, the final optimization procedure is formulated as:

$$\mathcal{L}^t = \lambda_{\text{topo}} \mathcal{L}_{\text{topo}}^t + \lambda_{\text{kin}} \mathcal{L}_{\text{kin}}^t, \tag{9}$$

where $\lambda_{\text{topo}}$ and $\lambda_{\text{kin}}$ determine the strength of regularization.

### 4.3 GENERATION VIA CONTROLLING $\mathcal{V}$

The proposed motion transfer framework can be naturally extended to support controllable video generation through manipulation of the velocity field $\mathcal{V}$. Since our videos are rendered from explicit 3D Gaussians, camera poses can be freely modified to generate videos from different viewpoints. Moreover, because the velocity field is also explicitly represented, it can be linearly transformed to alter the motion magnitude (speed) and direction. This enables fine-grained customization of the object behavior. Additionally, there is no inherent constraint on the number of frames in the output video, as the velocity field can be duplicated and manipulated arbitrarily. This allows it to be applied to the target object at any point in time, facilitating the production of extended and continuous motion. Therefore, Motion Marionette enables the generation of videos with arbitrary temporal lengths, diverse motion styles, and dynamic camera trajectories.

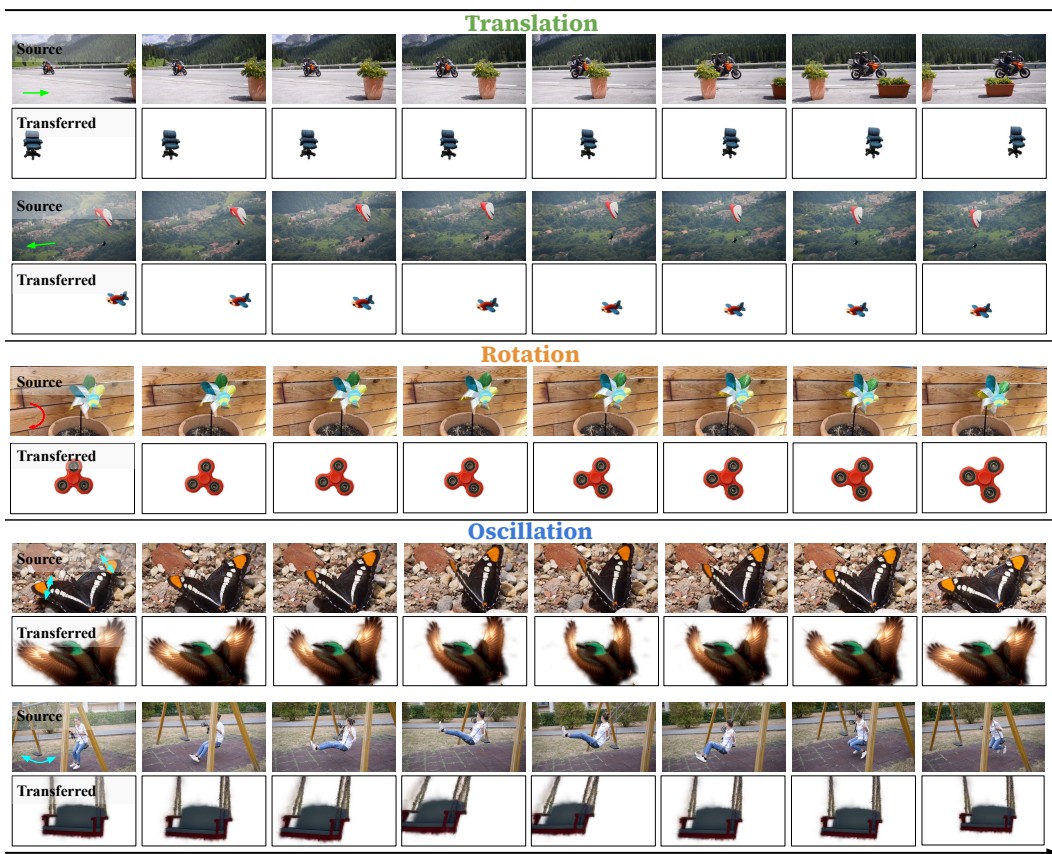

Figure 2: **Qualitative motion transfer results.** Time progresses from left to right. Arrows in the leftmost column indicate the approximate motion direction in the source video.

## 5 EXPERIMENTS

In this section, we conduct comprehensive experiments to investigate the following key questions:

- How accurate is the extracted SpaT prior from videos?
- How is the transferred video performance when integrating the SpaT prior with target objects?
- Can Motion Marionette be used for controllable video generation?

### 5.1 EXPERIMENTAL PROTOCOL

We conduct experiments using both real-world and synthetic data. For real-world data, we use monocular videos from (Gao et al., 2022; Pont-Tuset et al., 2018) to extract 3 distinct types of rigid motion, including translation, rotation, and oscillation, covering 8-10 unique dynamics. Each video features a clearly defined foreground object and distinct motion dynamics. To create diverse target scenarios, we use both Internet and synthesized static images for transfer performance evaluation, including three for translation, three for rotation, and four for oscillation. The target images typically have a resolution of $512 \times 512$ or $1024 \times 1024$ and depict photorealistic objects. For numerical comparisons, we adopt both reconstruction metrics (PSNR, SSIM (Wang et al., 2004), and LPIPS (Zhang et al., 2018)) and generation measurements (VideoScore (He et al., 2024)). We typically set $\lambda_{\text{topo}}$ and $\lambda_{\text{kin}}$ to 1. All videos are rendered using a fixed camera and a white background. Experiments are conducted on a single NVIDIA RTX A6000 GPU with additional implementation details provided in Sec. B.

### 5.2 SPAT PRIOR EVALUATION

To evaluate the effectiveness of different priors, we use the first frame from the source video as the target reference and assess how closely the transferred video matches the original one. Tab. 1

Table 1: **SpaT prior quality evaluation.**

| Motion Type | Prior | PSNR↑ | SSIM↑ | LPIPS↓ |
|---|---|---|---|---|
| Single-Slow | Gen | 17.77 | 0.40 | 0.57 |
| | SpaT | **19.08** | **0.95** | **0.09** |
| Single-Fast | Gen | 14.88 | 0.42 | 0.56 |
| | SpaT | **25.88** | **0.98** | **0.02** |
| Multi-All | Gen | 10.83 | 0.11 | 0.67 |
| | SpaT | **12.14** | **0.72** | **0.30** |

Table 2: **Efficiency comparison.**

| Method | Latency (min) |
|---|---|
| Trajectory Extraction | 6.5 |
| +SpaT Construction | 18.1 |
| +Transfer & Generation | 2.2 |
| **Total (Ours)** | **26.8** |
| DMT (Yatim et al., 2023) | 29.6 |
| VGM (WanTeam et al., 2025) | 38.7 |

compares the SpaT prior with the generative prior across three motion types: slow single-direction, fast single-direction, and multi-directional. This evaluation focuses on how accurately each prior reproduces the original motion sequence. Results show that our method consistently outperforms the generative prior, achieving better preservation of structural details and higher perceptual similarity. The geometric and simulation priors are excluded due to their limited applicability for generalizing across diverse videos and objects.

## 5.3 VIDEO TRANSFER PERFORMANCE

**Qualitative Comparison:** Fig. 2 presents visual results comparing the source and transferred videos. The target objects are segmented from their backgrounds to facilitate better comparison. Across all three motion types (translation, rotation, and oscillation), Motion Marionette successfully synthesizes video sequences that closely follow the dynamics (direction and speed) of the source videos, demonstrating the effectiveness of the proposed components. While needle-like artifacts are occasionally observed, this stems from the limitations of reconstructing accurate 3DGS representations from single-view images, which is not the focus of this paper.

We also compare Motion Marionette with representative methods using generative priors and simulation priors in Fig. 5. The baselines include Diffusion Motion Transfer (DMT) (Yatim et al., 2023), which leverages video generative priors, and PhysGaussian (Xie et al., 2023), which simulates motion through physics-based modeling. Motion Marionette preserves rigid object geometry more effectively than baseline methods over different motion types. While DMT is capable of generating detailed visuals, it lacks dynamic coherence and temporal consistency. PhysGaussian does not incorporate video-based guidance and relies entirely on predefined simulation parameters. The absence of external motion supervision often leads to unrealistic behaviors. As highlighted in the boxed regions, the propeller exhibits unintended deformations occur during rotation.

**Quantitative Comparison:** Due to the absence of ground-truth for transferred videos, a well-defined and universally accepted evaluation metric remains unavailable. For fair comparison, we use a third-party toolkit, VideoScore (He et al., 2024), to assess motion similarity between the source and transferred videos. VideoScore is a benchmarking tool designed to evaluate video quality across multiple dimensions, of which we focus on temporal consistency that measures the smoothness and stability of motion over time, and motion dynamics that assesses the degree of dynamic variation within the video. We apply

Table 3: **Video visual quality evaluation.**

| *VideoScore* ↑ | Translation | Rotation | Oscillation |
|---|---|---|---|
| DMT (TeS) | 0.79 | 0.97 | 0.78 |
| **Ours** (TeS) | 0.73 | 0.98 | 0.85 |
| DMT (DyS) | 0.85 | 0.98 | 0.97 |
| **Ours** (DyS) | 0.77 | 0.99 | 0.93 |
| *User Study* ↑ | Translation | Rotation | Oscillation |
| DMT (TeS) | 0.81 | 0.28 | 0.82 |
| **Ours** (TeS) | 0.93 | 0.91 | 0.85 |
| DMT (DyS) | 0.92 | 0.07 | 0.64 |
| **Ours** (DyS) | 0.94 | 0.82 | 0.72 |

VideoScore to both the source and transferred videos after background removal. To quantify similarity, we divide the score of the transferred video by that of the corresponding source video. This yields two metrics: **temporal similarity (TeS)** and **dynamic similarity (DyS)**, which together reflect how closely the motion patterns in the transferred video align with those in the source. We also conducted user studies to evaluate the transferred videos along the same two dimensions, where participants were asked to rate the similarity between source and transferred video pairs. More implementation details are provided in Sec. B.2.

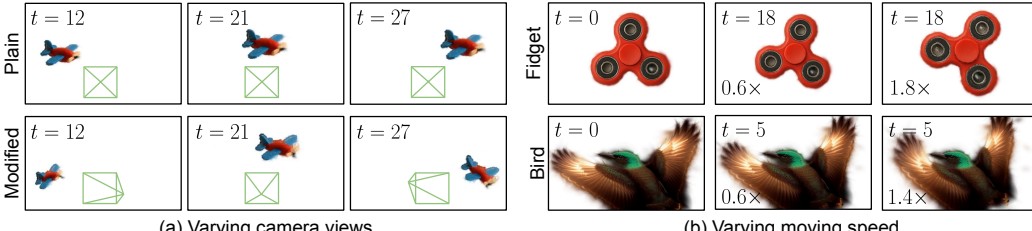

Figure 3: **Examples of controllable video generation.** (a) shows control over camera poses for generating different views; (b) shows results of varying motion speed through velocity scaling.

Tab. 3 presents the quantitative comparison results, with scores ranging from 0 to 1. Diffusion Motion Transfer (DMT) is evaluated against with, as it demonstrates the ability to generalize across diverse object categories, making it the most comparable baseline to our method. The results indicate that our method performs comparably to DMT under VideoScore evaluation, but significantly outperforms it in human assessments. We attribute this to the inherent spatial smoothness provided by the approximate yet structured nature of the velocity field in Motion Marionette, which contributes to enhanced perceptual coherence. Additionally, we observe that DMT is highly sensitive to the quality of text prompts. In cases where textual guidance is vague or underspecified, particularly in cases involving rotational motion, it frequently leads to corrupted video outputs.

### 5.4 CONTROLLABLE VIDEO GENERATION

The explicit object and velocity field representations in Motion Marionette enable flexible generation of an arbitrary number of video variants under different control parameters. As illustrated in Fig. 3, we demonstrate that our method allows users to control both camera viewpoints and object motion speed, while maintaining visual coherence. In Fig. 3(a), as the camera pose changes in conjunction with the object's movement, the object geometry remains consistent, resulting in a novel movement trajectory. In Fig. 3(b), scaling the motion speed produces temporally varied sequences while preserving geometric consistency.

### 5.5 ABLATIONS AND ANALYSIS

**Different Loss Effects:** In Fig. 4, we perform ablation studies to examine the impact of the loss terms defined in Eq. 7 (kinematic loss) and Eq. 8 (topological loss). Comparisons are made using video frames sampled at the same timestep for visual clarity. The kinematic loss primarily targets intra-object artifacts that occur within a single motion trajectory. It enhances local rigidity by preserving finer semantic details and reducing structural separations. In contrast, the geometric loss addresses inter-component inconsistencies by mitigating the disconnectivity between independently moving parts, thereby promoting smoother local geometry and producing a more unified and coherent object movement. A video comparison is also available on the anonymous website.

**Computational Efficiency:** We also evaluate and compare the computational efficiency of Motion Marionette with generative baselines (Yatim et al., 2023; WanTeam et al., 2025) in Tab. 2, where our method requires less time to obtain a transferred/generated video. For a 50 frame source video at a resolution of $854 \times 480$, gaining the final transferred video takes less than half an hour. After the SpaT prior is extracted, it can be applied onto any target object, which would only cost 2 minutes. These results highlight the efficiency of our pipeline and its suitability for scalable video generation.

## 6 CONCLUSION

We presented Motion Marionette, a novel zero-shot framework for rigid motion transfer from monocular videos to static images. By introducing the spatial-temporal (SpaT) prior as a generalizable and position-independent representation of motion, our method circumvents the limitations of prior-driven approaches. Through 3D Gaussian Splatting, explicit velocity field construction, and Position-Based Dynamics, Motion Marionette enables coherent video generation without relying on parametric models, simulations, or generative priors. Empirical studies confirm its effectiveness and flexibility in generating controllable motion sequences across different object types.

## ETHICS STATEMENT

The work focuses on developing a new perspective for motion transfer by framing it as a timewise spatial mapping between different objects. No personal data or sensitive information were involved in this study. The tasks and benchmarks do not raise ethical concerns related to safety, privacy, or misuse. We believe our work fully adheres to the ethical standards and guidelines of the community.

## REPRODUCIBILITY STATEMENT

To ensure reproducibility, we provide a detailed pipeline illustration in Fig. 1, outlining each step of the proposed method. In the methodology and experimental sections, we clearly specify the datasets, training protocols, evaluation metrics, and implementation details, including inference settings. Hyperparameters and pretrained models are reported to allow faithful replication of our results. In addition, source code will be released upon request to further support verification and reuse by the community.

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

## A  VIDEO DEMONSTRATIONS

We provide the synthesized videos in this anonymous project page.

## B  EXPERIMENT DETAILS

### B.1  IMPLEMENTATION DETAILS

The selected source videos in our dataset vary in resolution and frame count to better simulate real-world scenarios. The target images also differ in resolution, typically at 512×512 or 1024×1024. In general, higher-resolution images tend to yield better visual quality in the generated videos. For learning the 3DGS representations and motion trajectories across time steps, we follow the standard implementation provided in MoSca (Lei et al., 2024), and the maximum reconstruction optimization iteration is set to 6,000. During the velocity field learning stage, we update only the positions of the ellipsoidal Gaussians, while keeping their rotation, scale, opacity, and spherical harmonics parameters fixed. For the compared baseline methods, we follow their official repositories for both result reproduction and adaptation to our dataset. During the motion transfer process, peak GPU memory usage can reach up to 46 GB, particularly when processing high-resolution videos with long durations. All experiments are conducted on a single NVIDIA RTX A6000 GPU.

For local connectivity graph construction, we query up to 2048 nearest neighbors and employ a GPU-friendly implementation to accelerate computation. The number of Jacobi sweeps is set to 5, balancing local smoothness with optimization efficiency. The velocity threshold $\epsilon$ for identifying static regions is set to $1e^{-5}$. For simplicity, the weighting factors $\lambda_{\text{topo}}$ and $\lambda_{\text{kin}}$ are typically set to 1 in all experiments.

### B.2  USER STUDY DETAILS

We recruited 20 volunteers, all of whom are current or former graduate students, to evaluate the similarity of motion between the source video and the transferred video. In each trial, participants were presented with three videos: the source video, a video generated by Diffusion Motion Transfer (DMT), and a video produced by our method. Participants were asked to rate each video along two dimensions: temporal consistency ("Does the video appear authentic and smooth over time?") and dynamic degree ("Does the object exhibit clear and plausible motion?"). To standardize responses, the scores for DMT and our method were normalized by dividing them by the corresponding score for the source video, resulting in final scores ranging from 0 to 1.

### B.3  LIMITATIONS AND FAILURE CASES

Motion Marionette exhibits several limitations worth noting. Specifically, the visual quality of the generated videos is limited by the fidelity of the 3DGS reconstructed from single-view images. Additionally, the motion trajectories extracted from monocular videos may be noisy or inaccurate, particularly in complex, unconstrained real-world scenes. These challenges may be alleviated through future advancements in 3D and 4D reconstruction techniques.

Accordingly, we categorize the failure cases into two types. The first type involves artifacts that persist despite our refinement strategies, primarily caused by significant geometric discrepancies between source and target objects or inaccuracies in the extracted motion dynamics. The second type involves slight internal rotations that cause the object to gradually appear "decomposed" across time steps, which results from the inherently "thin" 3DGS representation produced by single-view reconstruction.

## C  ADDITIONAL RESULTS

In Fig. 4, we show the effectiveness of the proposed losses. While in Fig. 5, we compare Motion Marionette with methods using generative prior and simulation prior, showcasing the different characteristics of the methods.

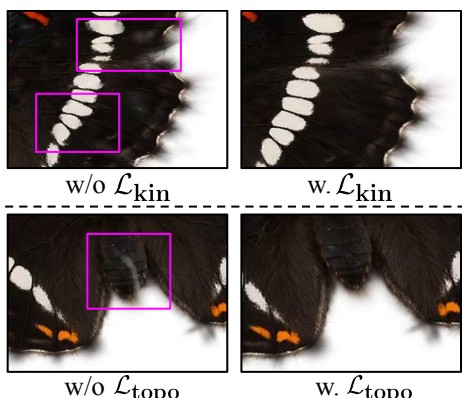

$$\text{w/o } \mathcal{L}_{\text{kin}} \qquad \text{w. } \mathcal{L}_{\text{kin}}$$

$$\text{w/o } \mathcal{L}_{\text{topo}} \qquad \text{w. } \mathcal{L}_{\text{topo}}$$

Figure 4: **Effect of the adopted losses.**

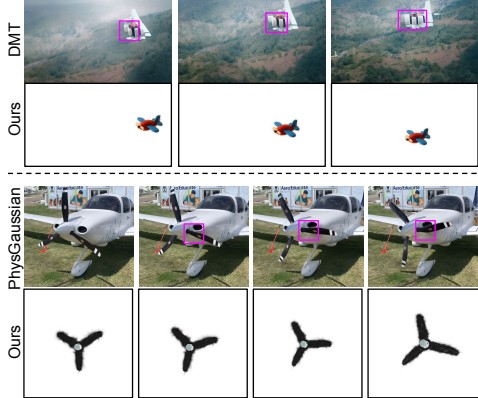

Figure 5: **Visual comparison illustration.** Boxes indicate the areas where artifacts arise.

## D  BROADER IMPACTS

Our work offers a new perspective on motion transfer by framing it as a timewise spatial mapping between different objects, which may provide deeper insights into the underlying mechanisms of motion adaptation. We hope this work can open a new research direction for motion transfer and controllable video generation. This direction has the potential to benefit a range of applications, including content creation, robotic manipulation, and safety-critical scene simulation. At present, we have not identified any potential negative impacts associated with this work.

## E  LLM USAGE

We used LLM (ChatGPT) to assist with writing refinement. Specifically, it was employed to improve clarity, grammar, and flow of text, as well as to adjust tone for academic writing. No content generation, experimental design, or analysis was delegated to the LLM; all technical contributions, mathematical derivations, and experimental results were developed by the authors. The LLM's role was limited to language polishing and presentation, and all outputs were carefully reviewed and edited by the authors.

