# OpenReview forum: "Motion Marionette: Rethinking Rigid Motion Transfer via Prior Guidance"
_ICLR.cc/2026/Conference — ICLR 2026 Conference Withdrawn Submission_

### Official Review · Reviewer_JTYN · 2025-10-31

**Soundness:** 3
**Presentation:** 3
**Contribution:** 2
**Rating:** 4
**Confidence:** 4

**Summary:**

This paper presents Motion Marionette, a novel framework for zero-shot rigid motion transfer from a source monocular video to a single static image. The method addresses the transfer of rigid motions such as translation, rotation, and oscillation. The core contribution lies in its departure from conventional approaches that rely on external geometric, generative, or simulation priors. Instead, the authors introduce an internal prior, which they term the Spatial-temporal (SpaT) Prior. This SpaT prior is designed to exclusively capture the spatial transformations over time, making the motion representation independent of specific object geometry and semantics, thus aiming for greater generalizability and temporal consistency.

**Strengths:**

1.	The paper proposes a clear and principled method for constructing the Spatial-temporal (SpaT) prior. It effectively extracts 3D motion trajectories from the source video and distills them into a sequence of rigid transformations (rotation and translation) using the Umeyama algorithm.
2.	This paper proposes a robust motion transfer process. It begins by applying the SpaT prior to the target's 3DGS representation to derive a velocity field. A crucial refinement stage is then introduced, featuring three optimization strategies inspired by Position-Based Dynamics. This stage is designed to effectively mitigate artifacts and enhance the temporal coherence and plausibility of the final animation.

**Weaknesses:**

1.	The framework's performance is fundamentally capped by the quality of its inputs. The process of reconstructing 3D representations from a monocular video and a single target image is inherently error-prone. These initial inaccuracies in 3D geometry can propagate through the pipeline, compromising the visual quality and stability of the final rendered video.
2.	The translation examples shown in Figure 2 are overly simplistic. For such straightforward movements, a much simpler baseline (e.g., using SAM for segmentation and then tracking the object's centroid) might suffice. The paper fails to clearly demonstrate the advantage of its more complex approach in these scenarios.
3.	Limited Capability in Handling Complex 3D Motion: The experiments predominantly feature motions that are planar. The method's ability to handle true 3D transformations is questionable. For source videos that include changes in 3D perspective (e.g., Fig2 Motorcycle and paraglider examples), the transferred results do not seem to replicate this crucial aspect of the motion. The examples demonstrating rotation are also overly simplistic, only involving rotation within a plane. When it comes to three-dimensional rotation, is this method still effective?
4.	Lack of Detail on Controllability: Section 4.3 claims the method supports controllable video generation, but it lacks sufficient implementation details, especially regarding how complex 3D transformations and dynamic viewpoint changes are practically achieved.
5.	Many of the resulting videos exhibit noticeable jitter and lack smoothness.
6.	The scope of the experiments is too narrow. The limited number of cases presented is insufficient to convincingly demonstrate the method's generalizability and effectiveness across a diverse range of objects and more complex motions.

**Questions:**

See Weakness

---

### Official Review · Reviewer_4UFV · 2025-11-01

**Soundness:** 3
**Presentation:** 3
**Contribution:** 1
**Rating:** 4
**Confidence:** 3

**Summary:**

This paper introduces Motion Marionette, a framework for transferring rigid motion from a source video to a target image. The core idea is to extract a geometry-independent "SpaT prior" from the source motion and apply it to a 3DGS representation of the target, generating a new video.

**Strengths:**

- The core idea of creating an "internal" spatial-temporal prior that only captures motion is good.
- Because the motion is represented as an explicit velocity field, the method allows for easy control over motion speed, camera view, and video length. The reported efficiency (under 3 minutes for transfer after prior extraction) is a strong practical advantage over slower generative models.

**Weaknesses:**

- The paper compares itself to diffusion models (DMT) and physics simulation (PhysGaussian), but it completely ignores the large body of work on flow-based or trajectory-conditioned video generation (like https://motion-prompting.github.io/). Methods that use optical flow or motion trajectories as guidance are highly relevant competitors, and not comparing against them makes the claim of a "new paradigm" feel overstated.
- The paper doesn't give a good reason for using 3DGS. The motion is modeled as a global rigid transformation, which could be applied to a much simpler representation like a point cloud or even directly in 2D image space via warping. Lifting to 3DGS from a single image is difficult and often produces flat, artifact-prone results, so using such a complex tool for this task needs a much stronger justification than what is provided.
- The SpaT prior is defined as a single rotation and translation per frame. This works for a single rigid object, but the paper claims to handle "oscillation" and shows a butterfly example, which involves multiple parts moving independently. The method section is very vague on how it handles multiple rigid motions; it seems the proposed formulation cannot support this without significant, unexplained extensions.
- The use of a normalized "VideoScore" is an unconventional and indirect way to measure performance. It's not clear what this "temporal similarity" score actually captures. The user study is also very small (20 participants) and may not be reliable. The evaluation lacks standard metrics and feels designed to favor the proposed method.

**Questions:**

Could you please clarify how the SpaT prior, which is formulated as a single rigid transformation per frame, can represent and transfer motion for objects with multiple independently moving parts (like the wings of a butterfly or bird)? The current description seems to only support a single, global rigid motion

---

### Official Review · Reviewer_uDZn · 2025-11-01

**Soundness:** 2
**Presentation:** 3
**Contribution:** 1
**Rating:** 2
**Confidence:** 5

**Summary:**

The paper proposes a zero‑shot rigid motion transfer framework that extracts a spatial‑temporal prior (SpaT) from a monocular source video and applies it to a single target image for motion-transferred video generation. The idea is to lift both source and target into 3D Gaussian Splatting (3DGS), then SpaT prior captures relative transforms (rotation/translation) between frames after Umeyama alignment. The target’s 3DGS is then driven by an explicit velocity field and refined using position‑based dynamics (PBD). The goal is to achieve controllable (viewpoint and motion scaling) video generation that is temporally consistent.

**Strengths:**

1. The paper is well written and easy to follow.
2. The internal‑prior formulation (SpaT) is category/semantics‑agnostic given it's leveraging multiple generalized approaches.
3. The use of 3DGS provide explicit control for objects.

**Weaknesses:**

1. The method’s core assumption of rigid motion significantly restricts its scope if the goal is to achieve motion-transferred video generation. Non‑rigid (cloth, humans) are largely out of scope but plays a big role in real-life videos.
2. The outputs are predominantly object-centric, whereas most competing baselines are designed for full-scene motion transfer. This mismatch in focus reduces the fairness and interpretability of quantitative and qualitative comparisons.
3. The choice of 3DGS as the core representation introduces inherent limitations: visible boundary artifacts appear in most results, and the approach cannot faithfully reconstruct or reason about occluded regions in the target image. These constraints question whether 3DGS is an appropriate representation for dynamic, scene-level video synthesis.
4. Dataset scale & diversity are modest (8–10 dynamics); more exhaustive studies and evaluations are required.

**Questions:**

I would hope the authors to address the weaknesses I've raised.

---

### Note · Authors · 2025-11-13

**Comment:**

We would like to sincerely thank the reviewers for their thoughtful and constructive feedback. We deeply appreciate the time and effort invested in evaluating our submission.

After reflecting on the comments and revisiting the work, we recognize that this paper is an *early exploration* of using explicit representations for general motion transfer under strict conditions (single video to single image). While the idea may hold promise, the current performance suffers from artifacts introduced during the 2D–3D transformation, which further raises concern about the suitability of 3DGS as the core representation. Although these components are not the main focus of our contribution, our attempt nevertheless reveals the inherent limitations of these methods under the given task constraints.

We are grateful for the reviewers’ insights, which have been extremely helpful in clarifying these limitations and guiding our next steps. Thank you again for the constructive engagement and suggestions.

**Withdrawal Confirmation:**

I have read and agree with the venue's withdrawal policy on behalf of myself and my co-authors.